# Using Machine Learning Algorithms to Develop a Clinical Decision-Making Tool for COVID-19 Inpatients

**DOI:** 10.3390/ijerph18126228

**Published:** 2021-06-09

**Authors:** Abhinav Vepa, Amer Saleem, Kambiz Rakhshan, Alireza Daneshkhah, Tabassom Sedighi, Shamarina Shohaimi, Amr Omar, Nader Salari, Omid Chatrabgoun, Diana Dharmaraj, Junaid Sami, Shital Parekh, Mohamed Ibrahim, Mohammed Raza, Poonam Kapila, Prithwiraj Chakrabarti

**Affiliations:** 1Milton Keynes University Hospital, Standing Way, Eaglestone, Milton Keynes MK6 5LD, UK; Vepa.abhinav@nhs.net (A.V.); Amer.Saleem@mkuh.nhs.uk (A.S.); Amr.Omar@mkuh.nhs.uk (A.O.); Diana.Dharmaraj@mkuh.nhs.uk (D.D.); MianJunaid.sami@mkuh.nhs.uk (J.S.); Shital.Parekh@mkuh.nhs.uk (S.P.); MohamedElbadri.Ibrahim@mkuh.nhs.uk (M.I.); Mohammed.Raza@mkuh.nhs.uk (M.R.); Poonam.Kapila@mkuh.nhs.uk (P.K.); Prithwiraj.Chakrabarti@mkuh.nhs.uk (P.C.); 2Leeds Sustainability Institute, Leeds Beckett University, Leeds LS1 3HE, UK; k.rakhshanbabanari@leedsbeckett.ac.uk; 3Research Centre for Computational Science and Mathematical Modelling, Coventry University, Coventry CV1 5FB, UK; 4Centre for Environment and Agricultural Informatics, Cranfield University, Bedfordshire MK43 0AL, UK; t.sedighi@cranfield.ac.uk; 5Department of Biology, Faculty of Science, University Putra Malaysia, Serdang, Selangor 43400, Malaysia; 6Department of Biostatistics, School of Health, Kermanshah University of Medical Sciences, Kermanshah 6715847141, Iran; n.salari@kums.ac.ir; 7Faculty of Mathematical Sciences & Statistics, Malayer University, Malayer 6571995863, Iran; o.chatrabgoun@malayeru.ac.ir

**Keywords:** Bayesian network, COVID-19, SARS CoV, random forest, risk stratification, synthetic minority oversampling technique (SMOTE)

## Abstract

Background: Within the UK, COVID-19 has contributed towards over 103,000 deaths. Although multiple risk factors for COVID-19 have been identified, using this data to improve clinical care has proven challenging. The main aim of this study is to develop a reliable, multivariable predictive model for COVID-19 in-patient outcomes, thus enabling risk-stratification and earlier clinical decision-making. Methods: Anonymised data consisting of 44 independent predictor variables from 355 adults diagnosed with COVID-19, at a UK hospital, was manually extracted from electronic patient records for retrospective, case–control analysis. Primary outcomes included inpatient mortality, required ventilatory support, and duration of inpatient treatment. Pulmonary embolism sequala was the only secondary outcome. After balancing data, key variables were feature selected for each outcome using random forests. Predictive models were then learned and constructed using Bayesian networks. Results: The proposed probabilistic models were able to predict, using feature selected risk factors, the probability of the mentioned outcomes. Overall, our findings demonstrate reliable, multivariable, quantitative predictive models for four outcomes, which utilise readily available clinical information for COVID-19 adult inpatients. Further research is required to externally validate our models and demonstrate their utility as risk stratification and clinical decision-making tools.

## 1. Introduction

On Thursday, the 5th of March 2020, within the UK, COVID-19 claimed the life of its first victim and has since contributed towards over 103,000 deaths [1,2]. The Office of National Statistics (ONS) has since issued statements, based on population data, in conjunction with the National Health Service (NHS), indicating an increased risk of mortality from COVID-19 amongst poorer socioeconomic groups, Black and Minority Ethnics (BAME), males and the elderly [2,3,4,5]. In addition to demographics, various biochemical parameters and co-morbidities, such as obesity, diabetes, hypertension, chronic obstructive pulmonary disease (COPD) and malignancy, have been identified as risk factors for poor COVID-19 outcomes [6,7,8,9]. However, using this vast data to improve clinical care has proven challenging. One particular challenge that remains is relatively quantifying the impact of various prognostic indicators upon COVID-19 outcomes, especially whilst using combinations of variables, in order to assist clinical decision-making and risk stratification.

Due to the limited nature of healthcare resources, such as hospital beds and ventilators, clinicians are often faced with difficult decisions where they must ration resources between patients, often having ethical implications [10,11,12]. Currently, clinicians are allocating healthcare resources to COVID-19 patients semi-quantitatively, and often as a response to clinical deterioration. Various risk stratification models have been described in the literature such as the 4C tool [13], but are currently not being used clinically due to criticism in recent systematic reviews [14,15]. Some of the key problems with existing risk-stratification tools are unclear methodologies, the exclusion of patients diagnosed with COVID-19 using Computed Tomography (CT) imaging but with negative Real Time-Polymerase Chain Reaction (RT-PCR) nasopharyngeal swabs, small sample sizes, many patients not reaching a study outcome, automated data extraction relying on clinical coding and many studies only exploring inpatient mortality as a primary outcome. In addition, many predictive models have been developed using patient data from other parts of the world, which may not be generalizable to the UK population due to patient factors, hospital factors and virus factors. Finally, only a small selection of risk-stratification tools analysed a wide host of independent variables including vital observations, biochemical markers, demographics and co-morbidities. 

The multivariate predictive model showcased in this study uses Bayesian Networks (BNs), which have received increasing attention during the last two decades [16,17] for their efficacy in tackling challenging and complex problems whilst also aiding in making decisions under uncertainty [18]. The ever-increasing volumes of health data has created potential for developing new knowledge that could improve clinical practise and patient care. The BNs and other machine learning (ML) methods have been extensively utilised in a diverse range of health topics from genomics [19,20,21] to treatment selection, and outcome, prognosis and prediction [22]. A compelling advantage of BNs over other suitable data-driven methods is that they do not explicitly need massively large datasets. Furthermore, BNs can combine the elicited knowledge of experts in circumstances where data are limited, and still produce meaningful and accurate decision-support systems [23,24,25,26].

This paper seeks to develop a quantitative tool to aid risk-stratification, and earlier clinical decision-making for adult COVID-19 inpatients by benefiting from the properties of BNs, including making reliable predictions, and being robust in making decisions under various sources of uncertainties in data.

## 2. Materials and Methods

### 2.1. Study Design and Setting

This retrospective case–control study was conducted at Milton Keynes University Hospital (MKUH), which is a medium-sized, 550 bed, district general hospital in the United Kingdom. Data was collected during the routine clinical care of patients for auditing purposes, and upon receiving Health Research Authority (HRA) approval, the anonymous data was then also used for research purposes. The study aimed to follow the Transparent Reporting of a multivariable prediction model for the Individual Prediction of Diagnosis (TRIPOD) checklist [27] and was conducted according to a pre-defined study protocol.

### 2.2. Study Population

Adult patients diagnosed with positive RT-PCR nasopharyngeal swabs or CT scans with changes suggestive of COVID-19 [28], between 01/03/2020 (date of first COVID-19 patient diagnosis) and 22/04/2020 (date of initiating independent predictor variable data collection) at MKUH, were included in this study. Sixty-nine patients were excluded, which is shown below in Table 1, to produce a final *n* number of 355. The sample size was determined by using the maximum number of COVID-19 patients diagnosed during the study period.

### 2.3. Data Collection

The hospital Picture Archiving and Communication System (PACS) was searched to get the details of the CT scan reports of patients with suspected COVID-19 changes, from 01/03/2020 until 22/04/2020. Reports dictated by a consultant radiologist, and CT scan images where required, were screened for all patients who had changes suggestive of COVID-19 [28]. The radiologically positive cases were included in the study. A record of all the COVID-19 RT-PCR positive swabs was obtained from the microbiology department. After removing the duplicates, the CT positive and RT-PCR swab positive cases were populated to a Microsoft Excel spreadsheet. Further patient data from the hospital Electronic Patient Record System (EPR), was collected in accordance with data protection and Good Clinical Practise (GCP) guidelines, on a hospital computer, by a team of physicians. Specific instructions were issued to the team of physicians to use during the collection of data to ensure homogenous, standardised interpretation of data from EPR. Healthcare staff who had historically recorded patient information on the EPR during clinical assessment were, of course at the time, blind to the outcomes and hypotheses of this study. All data was checked for systematic error by at least 1 other physician. After data collection, data was fully anonymised.

### 2.4. Independent Predictor Variables

Independent predictor variables were selected for inclusion in this study a priori based on three criteria; (i) having a postulated role for influencing COVID-19 severity based on surrounding literature, (ii) values expected to be available for at least one-third of study participants and (iii) values being collected during the routine care of study participants. Forty-four independent predictor variables were used for analysis in this study, which are shown and defined in Appendix A. Patient characteristic information is shown in Appendix A.

### 2.5. Outcomes

All patients were either discharged or deceased and thus achieved all four outcomes; (i) inpatient mortality (IPD), (ii) duration of COVID-19 treatment (ADT), and (iii) maximum level of oxygen or ventilatory support during inpatient stay (MOoVS), which was divided into 4 categories: (A) requiring room air or non-high-flow oxygen, (B) requiring high-flow oxygen defined as using a venturi mask, (C) requiring non-invasive ventilation (continuous positive airway pressure (CPAP) or bi-level positive airway pressure (BiPAP)) and (D) requiring intubation. A new radiologically confirmed diagnosis of pulmonary embolism during inpatient stay (NCPE) was the fourth outcome. The outcomes were selected for this study a priori based on 3 criteria: (i) all patients will be able to reach one of the pre-defined study outcomes, (ii) the pre-defined study outcomes are representative of COVID-19 severity based on the surrounding literature at the time of study design and (iii) the pre-defined study outcomes will involve data collected in the routine care of study participants. The follow-up period was defined as 2 months to give all patients ample time to achieve a study outcome prior to outcome data collection. During the above-mentioned data collection time-period, if a patient was admitted more than once, and both times were for COVID-19 related reasons within 5 days of each other, this was counted as a failed discharge and thus 1 admission. Days spent in hospital for social reasons or alternative diagnoses prior to developing COVID-19 were subtracted from the duration of inpatient stay to derive the duration of inpatient treatment outcome. The day of COVID-19 clinical presentation was retrospectively determined by physicians during data collection, after careful analysis of the patient notes, in order to determine the first day during the inpatient stay where COVID-19 was diagnosed clinically, based on the full repertoire of available clinical information.

### 2.6. Data Analysis

#### 2.6.1. Missing Values

In this study, no outcome (or dependent) variable was missing, but there were several independent variables with high numbers of missing values. The biochemistry features, including max CRP levels at the different days were among the independent random variables with the highest percentages of missing values ranging from 40% to 70%. However, limited numbers of missing values could be observed on the rest of independent variables. Due to the high number of missing values in many of the independent random variables, the multiple imputation (MI) technique was selected as the most suitable technique to estimate the missing values in the dataset.

#### 2.6.2. Balancing Outcomes

After converting the continuous and multi-scale discrete independent and dependent random variables into the categorical ones using the discretisation method [29], and estimating the missing values using the MI technique [30,31], it was observed that the resulting outcome variables suffer from a considerable imbalance. The predictions derived from fitting a suitable statistical model to the imbalanced datasets, whereby one class is dominant, would be inherently biased towards the dominant class, thus decreasing the reliability of the predictions made by the models [32,33,34].

In this study, the authors used the Synthetic Minority Oversampling Technique (SMOTE) to overcome the imbalance in the dataset [35]. The advantage of this technique over other oversampling methods is that it decreases the imbalance in a dataset by synthetically creating new examples of the minority class, and not duplicating them [35,36]. The authors applied the SMOTE on the entire dataset, in concordance with the surrounding literature [37,38,39,40].

#### 2.6.3. Feature Selection

One essential stage in the development of predictive models using supervised Machine Learning (ML) techniques is feature selection, which includes identifying and choosing the best combination of independent variables in a dataset for efficient and optimum analysis of the problem at hand [32,33].

For its feature selection, this study adopts the recursive feature elimination (RFE) method, which is a backward variable selection wrapper technique [40]. For this purpose, the authors computed the RFE method in R (version 4.0.2) using the random forest (RF) function embedded in the Caret package [41,42]. In this study, the performance of the wrappers is assessed using k-fold Cross-Validation (k = 10), which repeats five times. The result of the feature selection using the RFE method for each of these responses is shown below in Table 2. In this table, the numbers against each variable for the corresponding response indicates the predictive importance of that factor. The RFE method is a multi-step process. Firstly, the dataset is randomly split into 70% training and 30% testing using the validation set approach [35]. Subsequently, using the training data, a predictive model containing all features is developed based on the random forest method. This model then ranks the features based on a measure of importance. The RFE method then eliminates the least important feature, develops a new model based on a smaller number of independent variables, and re-ranks the remaining predictors [40]. RFE identifies two parameters: the number of subsets to evaluate and the number of predictors in each of the subsets. For each subset, the process of eliminating the least-important features continues until it reaches a determined subset size. Eventually, RFE compares the predictive performance of all subsets and determines the best subset size with the best accuracy [40].

Figure 1 shows the performance of the RFE method based on the ranks of the variables. Table 2 and Figure 1, Figure 2, Figure 3 and Figure 4 both show that for each outcome, there is a specific combination of independent variables that produce the highest predictive performance among all other possible combination of variables for the selected outcome.

### 2.7. Bayesian Network Modelling

A Bayesian network (BN) is a probabilistic graphical model that is used to represent knowledge about an uncertain domain [43]. Applications of BN methods are found in a growing number of disciplines and policies [44]. BN learning consists of two general steps: (i) Finding Directed Acyclic Graphs (DAG), which illustrates the inter dependency between the variables/nodes and is denoted by G, and (ii) Finding Conditional Probability Tables (CPT) for each node given the values of its parents on the learned network structure G.

Finding the best DAG is the crucial step in BN design. Construction of a graph to describe a BN is commonly achieved based on probabilistic methods, which utilise databases of records [45], such as the search and score approach. In this approach, a search through the space of possible DAGs is performed to find the best DAG. The number of DAGs, f(*p*), as a function of the number of nodes, *p*, grows exponentially with *p* [46].

The BN structure learned from the data only for IPD based on the feature selected factors affecting IPD is shown in Figure 5. This network structure was learned from the completed data by evaluating the best model out of various score-based or constraints-based methods [47]. In particular, the BN shown in Figure 5 was selected by employing hill-climbing (or hc) algorithm and benchmarked with other suitable learning algorithms (e.g., Tabu Search or simply tabu) available in “bnlearn” library in R package. We then used the cross-validation, which is a standard way to obtain unbiased estimates of a model’s goodness of fit to select the best models out of the learned networks using the learning algorithms mentioned above. The cross-validation method used in this paper is 5-fold cross-validation that can be simply computed using “bn.cv” function in “bnlearn” package. This function provides us with log-likelihood loss, its standard deviation and BIC. Both “tabu” and “hc” algorithms suggested the network structure illustrated in Figure 5, as the best networks learn from data, with the same BIC.

From this model shown in Figure 5, it is evident that the way that several independent variables affect IPD is incorrect. Therefore, it was important to discuss the resulting BN model, illustrated in Figure 5 with the domain medical experts, and consequently revise this BN by considering experts’ opinions. The revised BN model learned based on the combination of data and expert opinions, whilst also validated using several model diagnostic algorithms, such as k-fold cross validation, is illustrated in Figure 6. The computed BIC metric for the network shown in Figure 6 (−4190.4, equivalent to 9.03 of log-likelihood loss value and 0.008 standard deviation of the loss) is smaller than BIC of the model shown in Figure 5, which was computed to be, −4000.3 (equivalent to 8.69 of log-likelihood loss value and 0.011 standard deviation of the loss).

In the BN model proposed for modelling IPD, the strength of the link, as well as the associated uncertainty, is captured using probabilities and statistical distributions, which are estimated or derived based on the observed data. Figure 7 shows the learned BN with the estimated marginal probabilities shown on each node. In this BN, three nodes (Age, Chronic Kidney Disease (CKD), and Ethnicity) are considered as root nodes, and their parameters are learned by estimating these probabilities using the maximum likelihood method or Bayes estimate. The estimated marginal and conditional probabilities for the variables can be updated in the light of new evidence or data using a statistical algorithm known as the Bayes rule [45]. Hence, the BN can compute the probability of surviving or dying due to COVID-19 based on the different combination of the parent nodes, including Age, the minimum Albumin level during admission (MADA), and the mean C-Reactive Protein (CRP) level during days 7–8 since clinical presentation of COVID-19 (MCRP7).

Using the same methods, BNs were constructed for the other 3 outcomes and are shown in Appendix A.

## 3. Results

### 3.1. Inpatient Mortality (IPD)

The results section describes the performance metrics of the predictive models constructed in this study, as well as the conditional probabilities of outcome occurrence given different exemplary combinations of independent factor variables. 

Based on the BN model shown in Figure 7, the probabilities of IPD and survival of COVID-19 inpatients at different age groups can be computed, as shown in Appendix A. It highlights that the death rate of COVID-19 patients with ≥70 years is five times larger than patients’ with ≤40 years. These probabilities can be updated by observing more evidence about the states of other influencing variables. Table 3 shows the conditional probabilities of IPD given MADA (1 = <30, 2 = 30–35, 3 = 35<), MCRP7 (1 = <50, 2 = 51–100, 3 = 100<), and Age in years (1 = <40, 2 = 40–70, 3 = 70<). It can be concluded that for the patients in the first age group (<40 years) if the MADA is above 35 g/L, they will survive COVID-19 regardless of MCRP7. For a patient in this age group and with the MADA level less than 30 g/L, they would more likely survive if their MCRP7 were less than 50 mg/L. This means that when the MADA level is less than 30 g/L, patients are at a particularly high risk, especially if their MCRP7 level is above 50 mg/L. Interestingly, similar patterns can be found for the patients aged > 70 years old, but the corresponding survival probabilities are considerably lower.

Overall, these results indicate how low albumin (reflective of malnourishment, as well as infection owing to its negative acute phase protein property), high CRP, and old age correlate with inpatient mortality in an additive manner.

Oxygen Saturations (OS) at the time of presentation with COVID-19 were also associated with mortality. As OS decreases from >92% to <92%, the risk of mortality increases by 2%, suggesting that there is a negative association between these two variables, as shown in Appendix A. We expected this association to be stronger, but found it difficult to measure the oxygen saturations during admission at any other time, or as an average, as it would be heavily affected by the level of oxygen therapy being received by the patient. Although these findings are useful, it is also fascinating to observe how OS could jointly, with other influencing independent variables, affect the risk of inpatient mortality, as shown below in Table 4, where OS in % (2 = <91, 1 = 92<), CCX (1 = ‘No’, 2 = ‘Yes’), Ethnicity (1 = ‘Caucasian’, 2 = ‘Non-Caucasian’), and Age in years (1 = <40, 3 = 70<). The illustrated results in this table suggest that the risk of inpatient mortality is elevated for patients with reduced oxygen saturations and older patients. Ethnicity seems to increase the risk of death in patients 70< years, which is concordant with surrounding literature [3]; however, the results were not true for younger patients. This may have been due to a skewed population demographic, whereby older patients tended to be Caucasian and younger patients reflected a more multicultural demographic. Changes on chest X-ray (CCX) did not seem to significantly affect the risk of IPD, perhaps because the presence of changes is more likely an indicator of the time-point that an individual is along in their COVID-19 infection rather than an indicator of severity.

The next important research question is how the trend in CRP levels during the clinical course of a COVID-19 infection can be incorporated and evaluated using an appropriate model. This is because CRP levels can often correlate with infection severity, with a small associated lag time. Therefore, the trend in CRP is clinically useful for predicting what will happen to the patient. For example, if the gradient between the latest two CRP variables were positive, it would indicate that the infection is getting worse, whereas if the gradient were negative, it would indicate infection resolution. To account for the gradient between the CRPs, a dynamic version of BN needs to be developed, which is not possible due to the lack of training data. However, the BN model, illustrated in Figure 7, can be used to compute the risk of inpatient mortality for different levels of CRP at the different days during the clinical course of COVID-19 infection, as shown below in Table 5. This table shows the conditional probability of IPD given different configurations of Mean CRP between days 1–2 since clinical COVID-19 presentation (MCRP1) (1 = <30, 2 = 31–100, 3 = 100<) and Mean CRP between days 7–8 since clinical COVID-19 presentation (MCRP7) (1 = <50, 2 = 50–100, 3 = 100<), Age in years (1 = <40, 3 = 70<) and Minimum Albumin During Admission (MADA) (1 = <30, 2 = 30–35, 3 = 35<). As shown, if the level of MADA is 35<, increases or decreases in CRP levels during the clinical course of COVID-19 infection will not impose a mortality risk in patients <40 years. However, in patients aged 70< years, any increase in CRP levels (mg/L) from days 1–2 to days 7–8, would significantly increase mortality risk.

Table 6 below indicates the promising Positive Predictive Value (PPV), Negative Predictive Value (NPV), Sensitivity, Specificity, Overall accuracy and F-Score, which are used to evaluate the predictive performance of the BN suggested to model IPD in Figure 7, in terms of the feature selected risk factors. The definitions and details of how these metrics can be computed are described in [35]. F1 Score is the Harmonic Mean between precision and recall. The F1 score (83.7%) and accuracy (84.1) of our BN model developed for IPD is high, despite the small dataset used to train and test our BN model. Eighty-two percent (PPV) of adult patients predicted to die as inpatients during clinical COVID-19 infection, by our model, will die. However, only 67.86% (NPV) of adult COVID-19 patients predicted to survive the inpatient admission will indeed survive. This indicates that our model may fail to predict inpatient death of a sub-set of adult COVID-19 patients, but we expect this to improve with a larger dataset, which also incorporates more variables such as socioeconomic factors.

### 3.2. Duration of Inpatient Treatment for COVID-19 (ADT)

Understanding the simultaneous impact of MADA, obesity, and MCRP1 on the duration of COVID-19 treatment in patients is increasingly important to manage the growing, unrelenting pressures on hospitals and the NHS. Table 7 shows the probabilities of several important queries computed from the learned BN. In this table, ADT categories are ‘1’ (<1 day), ‘2’ (>2 days but < 3 days) and ‘3’ (>3 days). MCRP1 has been divided into 3 categories: ‘1’ (<50), ‘2’ (51–100) and ‘3’ (>100), and MADA has been divided into 3 categories: ‘1’ (<30), ‘2’ (30–35) and ‘3’ (>35). As is evident from this table, the duration of treatment of 71% of non-obese COVID-19 patients with normal MADA levels (>35 g/L) and low MCRP1 (<30 mg/L) is up to 1 day. In addition, the treatment duration of 95% of the patients with the above characteristics would be less than 3 days. In comparison to the obese patients, it can be observed that this probability (i.e., probability that the duration of treatment is up to one day) will be reduced to 54.4% (with the same characteristics). On the other hand, the predicted probabilities for duration of treatment of the obese and non-obese patients, with very low levels of MADA (<30 g/L), regardless of levels of MCRP1, are not significantly different from each other. Furthermore, the model and results reported in Table 7 suggest that the levels of MCRP1 alone would not be adequate to accurately predict the probabilities of treatment duration of more than 3 days. These probabilities must be updated by adding more evidence about the levels of MCRP at other days. It would be straightforward to revise and update the BN by augmenting the other outcomes, for example “IPD”, to understand what proportion of patients with a treatment duration <3 days may not survive. Overall, it can be observed that the COVID-19 treatment duration is higher for obese patients with high CRP levels and low Albumin levels.

Several metrics to assess the predictive performance of the proposed BN for the three different categories of ADT are shown in Table 8. 

The computed sensitivity measures, which is the metric to evaluate the learned BN ability to predict true positives of each available category of ADT, suggest that “ADT > 3” days has the highest rate (77%), and ‘2 ≤ ADT < 3^’ days has the lowest sensitivity rate (41.4%). We also compute specificity, which is the metric to evaluate the fitted BN ability to predict true negatives of each ADT category. The results suggest “ADT < 1” days (83.5%) and “ADT > 3” days are the categories with the highest and lowest specificity rates, respectively. The next important metric is F1-score that can be interpreted as a weighted average of the precision and sensitivity values, where an F1 score reaches its best value at 1 and worst value at 0. Since, the F1-score takes both false positives and false negatives into account; it will be usually more useful than accuracy, especially if the original test dataset has an uneven class distribution. The computed F1-scores for the ADT categories suggest promising accuracy for ‘ADT < 1’ days and ‘ADT > 3’ days.

### 3.3. Max Oxygen or Ventilatory Support (MOoVS)

It is of great important to understand how the right level of oxygen therapy or ventilatory support should be selected to enhance the survival rate and recovery speed of COVID-19 patients. Table 9 shows the conditional probabilities (as heat-mapped) of Max Oxygen or Ventilatory Support (MOoVS) given the different configurations of MADA (1 = ‘<30’, 2 = ‘30–35’, 3 = ‘35<’), OS (‘1’ = >92 and ‘2’ < 92) and MCRP11 (1 = <100 and 2 = >100). 

Patients either required no high-flow oxygen (NHF), high-flow oxygen (HF), CPAP/NIV (CPN) or ITU admission (ITU). The illustrated results in this table suggest that most of the patients with the better health characteristics such as high MADA (>35 g/L), high OS (≥92%), and low MCRP11 (≤100 mg/L) are likely to only require no-high flow O2 (73%) or high flow O2 (12.5%). These patients are thus suitable for ward-based care and will not need ITU admission. On the contrary, when MADA and OS levels, respectively, drop to below 30 g/L and 91%, and MCRP11 level increases to over 100 mg/L, the need is increased for these patients to receive CPAP/NIV (37%) or ITU admission (49%).

The probabilities given in Table 9 can be significantly altered in the light of new evidence, such as patient age. The updated probabilities are shown below in Table 10. Table 10 shows the conditional probabilities (as heat-mapped,) of MOoVS given the different configurations of MADA, OS, MCRP11, and Age (1 = ‘<40 years’ and 3 = ‘>70 years’). As mentioned above, patients required either NHF, HF, CPN or ITU. 

From this table, it can be concluded that all young patients with OS ≥ 92, MADA ≤ 30 and MCRP11 ≤ 100 require either NHF (80%) or HF (20%) to recover. If MADA levels of young patients increase to over 35 g/L, with the same levels of OS (≥92) and MCRP11 (<30), they will more likely need to use only NHF (93%) to recover. However, if the health of the patients starts to deteriorate, as MADA and OS levels, respectively, drop to <30 g/L and <92%; and MCRP11 level increases to over 100 mg/L, their need for HF, CPAP/NIV (CPN) or ITU will significantly increase depending to the age of patient. For the young patients, CPAP/NIV (36.5%) or ITU (63.5%) would be recommended. However, for patients > 70 years, either HF (22%) or CPAP/NIV (76%), and paradoxically not ITU (0%), would be recommended, usually because they are deemed unsuitable for ITU admission due to the futility of ITU-based treatment relative to a younger patient. This reflects the rationing of healthcare resources that occurs in hospitals after difficult medico-ethical decisions.

Table 11 illustrates several metrics to assess predictive performance of the proposed BN for the different states of “MOoVS”. The overall classification accuracy suggests that over 60% of cases have been correctly classified. Despite the small sample size of data and high percentages of missing values of the raw data, the computed overall accuracy (60.25%) is quite promising. The ability to predict true positives of each available category of MOoVS is measured by Recall, or Sensitivity, which suggests ITU admission has the highest rate (89%), and “CPN” has the lowest sensitivity rate (36%). We also compute specificity, whereby “CPN” (91%) and ITU admission (73%) are the categories with the highest and lowest rates, respectively. Since the original raw data has an uneven class distribution, F1-score as a weighted average of the precision and sensitivity values were calculated.

### 3.4. New Confirmed Pulmonary Embolism during Admission (NCPE)

Understanding the influence of various independent variables upon the conditional probability of having an NCPE is important for clinicians in guiding their use of thromboprophylaxis within the context of COVID-19. Our results confirmed that having bilateral COVID-19 changes on CT scan can increase the risk of NCPE from 13.9% to 72.4%. This suggests that more extensive ground-glass or consolidative changes on CT scan secondary to COVID-19 may be associated with NCPE secondary to COVID-19. Another well-known key predictor variable is the maximum D-dimer during admission (MDD), which is often an indicator of thrombosis. Our results indicated that having an MDD of 400<, as opposed to <400, increases the risk of NCPE from 34.6% to 54.5%. Interestingly, NCPE appeared to be more significantly influenced by the presence of MADA < 30 or bilateral ground-glass or consolidative CT scan changes secondary to COVID-19, rather than levels of MDD, as shown below in Table 12. This may be explained by the fact that many other conditions can also increase D-Dimer, such as disseminated intravascular coagulation, deep vein thrombosis and infection, which could thus be leading to high rates of PE false positives amongst COVID-19 patients. Furthermore, not all patients with raised D-dimers would have had CT-scans to investigate for PE, especially if it was deemed futile and the patient was palliative.

We finally investigated the impact of obesity alongside MADA and MCRP7 levels on NCPE in Table 13. It can be concluded that there is a strong association (70%) between the presence of NCPE and MADA < 30, as well as MCRP7 > 100, for the non-obese patients. If the patient is obese, MADA < 30 seems to be more influential in contributing towards the risk of NCPE, as opposed to MCRP7 > 100.

Various metrics, including PPV, NPV, sensitivity, specificity, overall accuracy, and F-Score, have been used to evaluate the predictive performance of the BN created to model NCPE, as shown in Table 14. The computed F1-score of almost 86% shows the classification prediction of the learned BN for NCPE is precise and robust. In addition, the PPV (83.7%), sensitivity (88%) and NPV (80.9%), which collectively represent the BN model’s ability to predict inpatient NCPE, was high.

## 4. Discussion

In this study, in addition to quantifying the significance of feature-selected risk factors, we showcase the use of Bayesian Networks to accurately predict four different COVID-19 inpatient outcomes, using different combinations of readily available clinical data, which serve as the independent predictor variables, whilst also accounting for interdependency between these variables.

Various COVID-19 prognostic indicators have been described in the literature, such as neutrophil:lymphocyte ratio, CRP, age, gender, ethnicity, oxygen saturation on admission, diabetes mellitus, hypertension, malignancy, obesity and COPD [48]. However, drawing insights from this information is impeded by the lack of clarity as to the relative influence each of these indicators has on mortality. In the clinical setting, patients often present with different combinations of these risk factors and biomarkers. Consequently, weighing them all up to allocate scarce healthcare resources can be challenging. This highlights a role for a predictive, quantitative risk-stratification tool. Our model has been constructed so that it can utilise data at first clinical presentation, for example, in the emergency department, but also after 3 and 7 days of inpatient treatment. This can allow clinicians to risk stratify at different time-points during an inpatient stay. In addition, our model can also predict the duration of inpatient COVID-19 treatment and maximum level of oxygen requirement that a patient may need during their inpatient stay. This may aid emergency physicians with the decision as to whether to admit a patient to hospital and avoid failed discharges, but also medical physicians with the decision as to whether to refer to ITU prior to a patient’s clinical deterioration. The predicted duration of inpatient COVID-19 treatment can be especially useful for bed managers in orchestrating patient flow, which is essential to prevent the growing problem of hospital-acquired COVID-19 secondary cross-contamination [49]. Furthermore, within the context of hospitals, which have reached their maximum ITU capacity, by using our predictive model to identify high-risk patients earlier on in their disease course, clinicians can transfer these high-risk patients to neighbouring hospitals prior to their clinical deterioration. 

One of the major strengths of our study is that we predict four outcomes. This increases the amount of clinical utility that can be offered to guide clinical decision making. Secondly, this data set has incorporated 44 different variables from 355 patients who all received the primary study outcomes, reducing the risk of confounding errors and selection bias. Another strength of this study is that data extraction was conducted and checked manually by trained medical physicians rather than by using coding. It has been well documented that hospital clinical coding is still not entirely accurate within the UK [50,51], and therefore insights drawn from national databases may be prone to significant information bias and thus systematic error. Furthermore, it has now been estimated that the sensitivity of the COVID-19 RT-PCR nasopharyngeal swab is likely to be 50–75% [52,53,54,55,56]. This has created a huge global problem in diagnosing and identifying COVID-19 patients, especially because not all patients exhibit symptoms [57,58,59,60,61]. To overcome this issue, the cohort of patients with COVID-19 who had negative RT-PCR swabs but positive CT scan imaging (*n* = 55) were also included in this study. 

The biggest limitation to our study is that since the study was conducted in a single centre, not only is the *n* number limited, but also the data only reflects the demographics of the surrounding population. Within the UK, geographical location and socio-economic factors are heavily influencing death rates [3] and, thus, our data may have limited generalizability to the wider UK population by not accounting for these factors. The level of generalizability could have also been better assessed by using external validation methods to test the performance of the model, but this would have required an additional independent dataset. Secondly, data was not always available, or accurate, for all patients. This was sometimes due to a lack of documentation, usually if the attending physicians at the time did not deem the information relevant, or if the information was not available, especially in patients who were cognitively impaired without any next of kin to provide collateral histories. Moreover, not all investigations, such as CT scans, were required for every patient and may have not been done due to the limited resources available in the NHS. Subsequently, only patients deemed to have abnormal results would have been the patients to receive the investigation. Additionally, some patients had different treatment goals to others. For example, patients with severe COVID-19 who could have had ventilatory support may have not had it because their treatment goal was palliation instead. All these factors together introduce information bias secondary to data measurement. Finally, although most parameters were objectively documented, some data, such as ethnicity, was self-reported by patients, thus also introducing a modest element of recall bias.

As a future work, it would be appealing to further investigate the impact of SMOTH technique on the validity of the BN classifiers. As discussed above, it was observed that the response variables in this study are significantly imbalance (e.g., the minority class percentages of NCPE or MOoVS were less than 5%). In order to overcome the imbalance in the dataset, the SMOTE technique was applied on the entire dataset, as recommended in [62,63,64,65]. However, the alternative approach, which is more plausible, would be to apply the SMOTH on training datasets only, and re-evaluate the performance measures using the test data.

## 5. Conclusions

In this study, we were facing several numerical challenges whilst constructing a robust, reliable and computationally efficient probabilistic data-driven model, including a very small sample size in comparison to the dimension of input variables, and significant rates of missing values for several variables. These challenges were all efficiently resolved by selecting and employing a range of ML methods. The output of this study was a quantitative tool, which can aid in both risk-stratification, and earlier clinical decision making, for COVID-19 inpatients.

Overall, our findings demonstrate reliable, ML-based predictive models for four outcomes that utilise readily available clinical information for COVID-19 adult inpatients. Our model not only computes the probability distributions of children nodes given the values of their parent nodes, but also the distributions of the parents given the values of their children. In other words, they can proceed not only from causes to consequences, but also deduce the probabilities of different causes given the consequences. All these probabilities can be instantly computed using the codes developed in R. The codes are available on request from the corresponding author (A. D).

The developed models in this study, if provided with more training data, have the potential to be refined even further. Future research is required to externally validate our models and demonstrate their utility as clinical decision-making tools.

## Figures and Tables

**Figure 1 ijerph-18-06228-f001:**
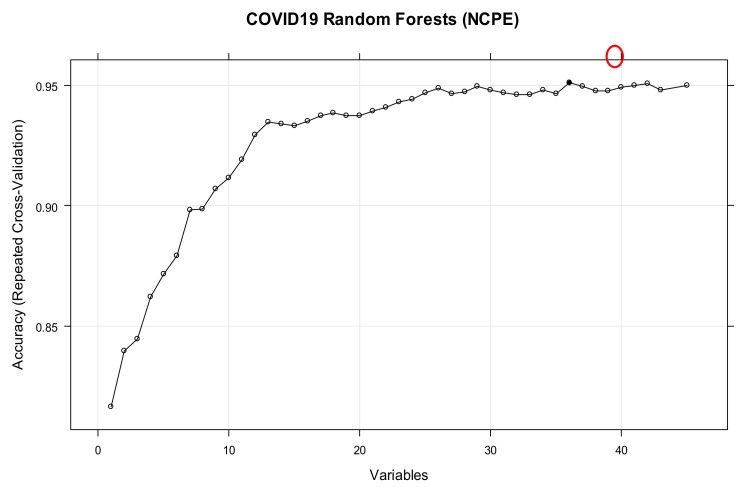
Performance of the RFE based on the ranks of the features of NCPE. The red circle shows the maximum achievable performance based on the best combination of variables.

**Figure 2 ijerph-18-06228-f002:**
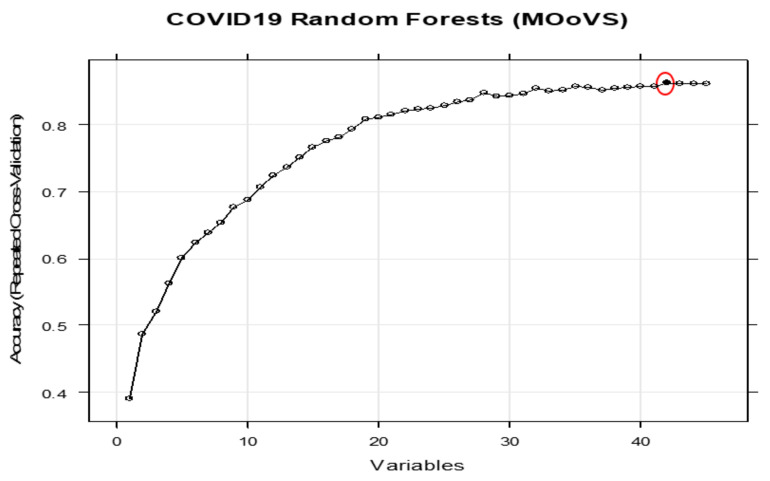
Performance of the RFE based on the ranks of the features of MOoVS. The red circle shows the maximum achievable performance based on the best combination of variables.

**Figure 3 ijerph-18-06228-f003:**
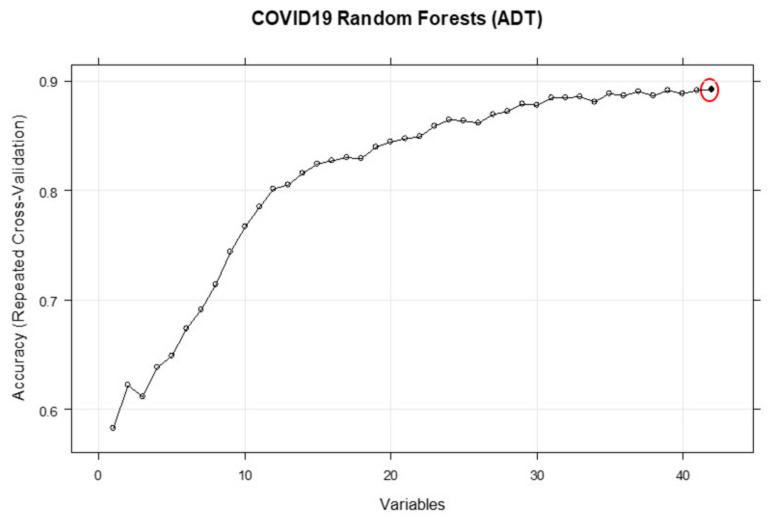
Performance of the RFE based on the ranks of the features of ADT. The red circle shows the maximum achievable performance based on the best combination of variables.

**Figure 4 ijerph-18-06228-f004:**
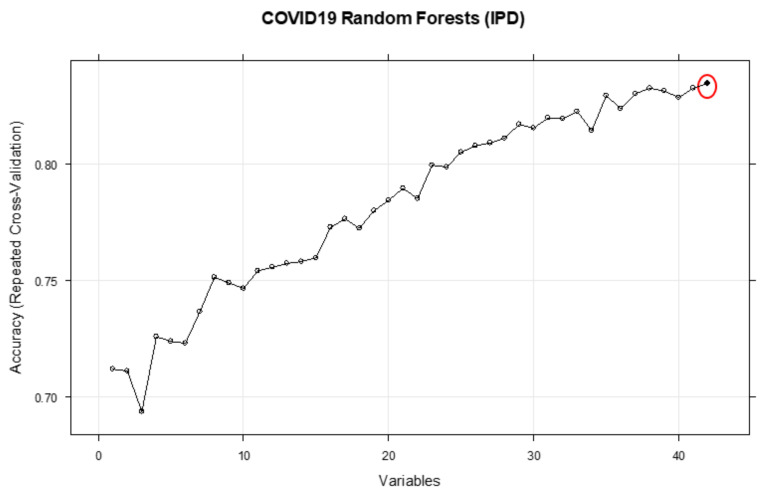
Performance of the RFE based on the ranks of the features of IPD. The red circle shows the maximum achievable performance based on the best combination of variables.

**Figure 5 ijerph-18-06228-f005:**
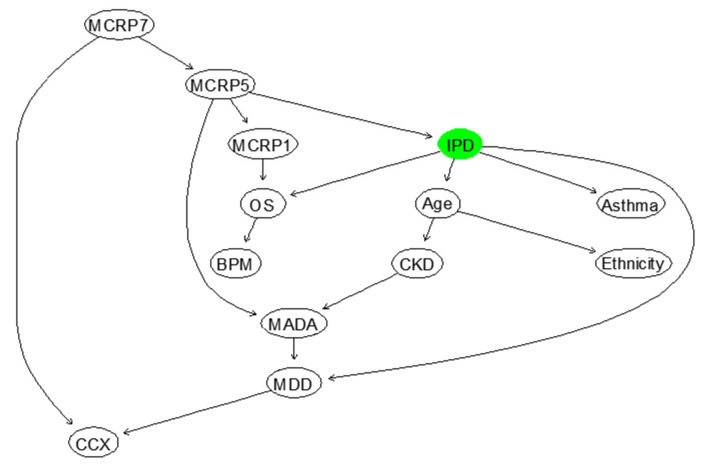
The BN that is fully learned from data to model “IPD” in terms of other relevant factors.

**Figure 6 ijerph-18-06228-f006:**
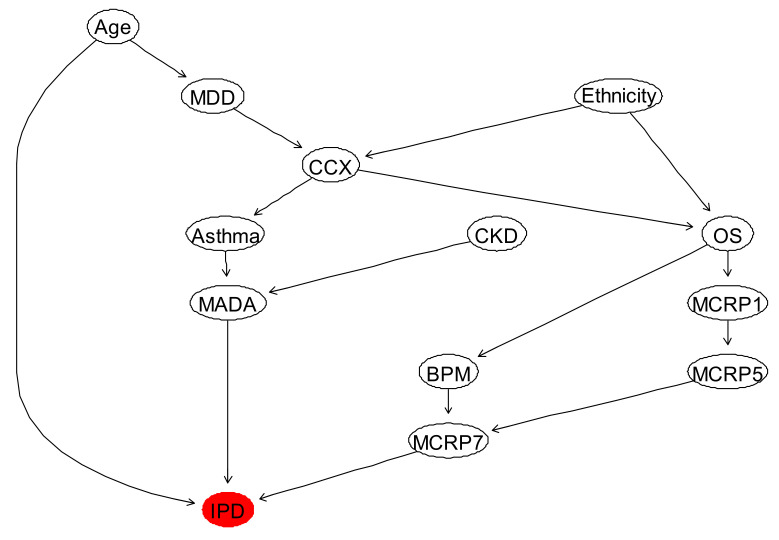
The BN learned by eliciting the domain expert combined with the (balanced and completed) data.

**Figure 7 ijerph-18-06228-f007:**
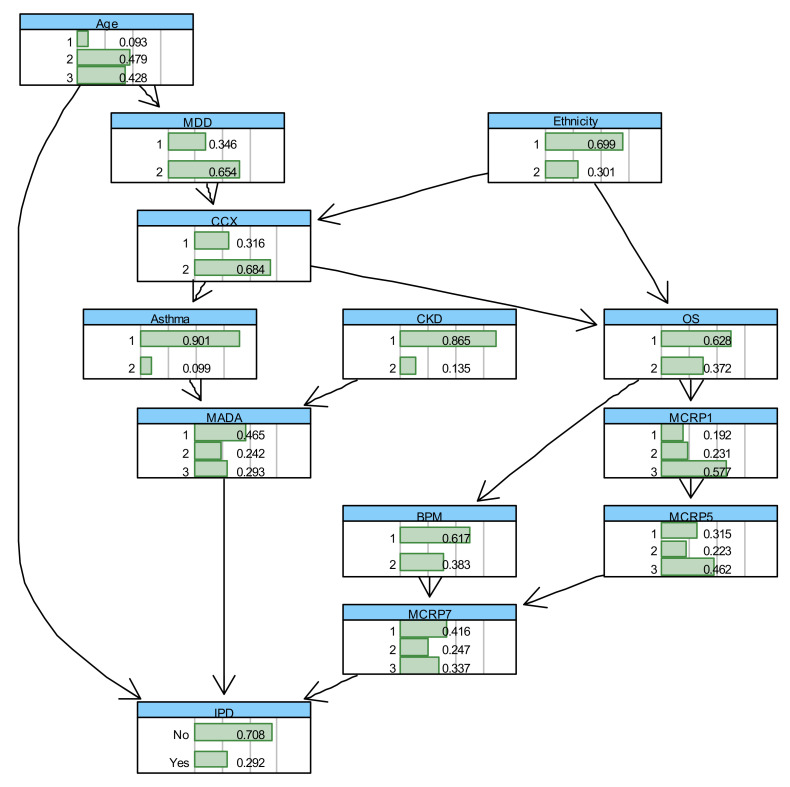
The BN with conditional probability tables (CPT) learned for “IPD” outcome based on the combined elicited domain expert opinions with the (balanced) data.

**Table 1 ijerph-18-06228-t001:** Patient selection process.

Sample Population	
Patients diagnosed with COVID-19 between 01/03/2020 and 22/04/2020, at Milton Keynes University Hospital (*n* = 424)	
Inclusion Criteria	Exclusion Criteria
Patients diagnosed with at least 1 positive RT-PCR Nasopharyngeal swab Patients diagnosed with CT scan changes consistent with COVID-19 [28] Age 18 years and above	Patients diagnosed in the Outpatient setting Staff Members who were diagnosed via Occupational Health, and who did not receive a formal medical assessment
Final Study Participant Number (*n* = 355)	

**Table 2 ijerph-18-06228-t002:** Feature selection results for four different outcomes; IPD, ADT, NCPE and MOoVS.

Predictor	RFE (NCPE)	RFE (MOoVS)	RFE (ADT)	RFE (IPD)
Age	2	1	7	1
Gender (Ge)	7	17	17	33
Ethnicity	17	5	24	4
Oxygen Saturations (OS)	16	2	9	2
Respiratory Rate (BPM)	19	12	26	9
Temperature	6	10	19	35
Obesity	11	8	11	28
Previous Venous Thromboembolism (PVTE)	Rejected	33	33	36
Chronic Obstructive Pulmonary Disease (COPD)	Rejected	37	37	37
Bronchiectasis	Rejected		41	31
Asthma	21	27	34	10
Interstitial Lung Disease (ILD)	Rejected	21	40	38
Lung Cancer (LC)	Rejected	41	38	39
Diabetes Mellitus (DM)	29	16	27	21
Hypertension (HTN)	14	26	8	25
Ischaemic Heart Disease (IHD)	31	28	28	16
Chronic Kidney Disease (CKD)	32	31	31	8
Non-steroidal anti-inflammatory drugs (ANNC)	33	38	25	15
Anticoagulant	23	35	29	17
Long-Term Antibiotic (LTA)	Rejected	34	36	30
Long Term Oral Steroid (LTO)	Rejected	39	42	42
Immunosuppressants (ISES)	Rejected	32	39	32
Oral NSAIDs (ONS)	Rejected	40	32	41
Angiotensin Converting Enzyme Inhibitors (ACEI)	28	36	30	40
Angiotensin Receptor Blockers (ARBB)	27	29	35	27
CT imaging severity of COVID-19 related changes (UoB)	1	4	2	23
COVID-19 related Chest X-ray changes (CCX)	30	7	10	11
Lactate (LDP)	12	25	21	20
Lymphocytes (LyDP)	4	23	16	18
Neutrophils (NDP)	5	18	15	26
Albumin (MADA)	3	6	1	6
Ferritin	24	20	23	24
D-Dimer (MDD)	8	11	6	7
C-Reactive Protein (CRP) Day 0	18	13	3	19
CRP Day 1–2 (MCRP1)	13	22	4	12
CRP Day 3–4 (MCRP3)	20	19	14	22
CRP Day 5–6 (MCRP5)	10	15	12	5
CRP Day 7–8 (MCRP7)	9	3	5	3
CRP Day 9–10 (MCRP9)	22	14	13	14
CRP Day 11–12 (MCRP11)	26	9	18	29
CRP Day 13–14 (MCRP13)	15	24	20	34
CRP Day 15–20 (MCRP15)	25	30	22	13

**Table 3 ijerph-18-06228-t003:** The conditional probability of IPD given different configurations of the parent nodes.

Risk Factor (MADA, MCRP7, Age)	Probability of Inpatient Mortality
(3, 1, 1)	0
(2, 1, 2)	0
(3, 3, 1)	0
(2, 2, 2)	0.20
(1, 1, 1)	0.33
(1, 3, 1)	0.397
(3, 1, 3)	0.417
(1, 1, 3)	0.513
(1, 2, 2)	0.594
(3, 3, 3)	0.813
(1, 3, 3)	0.866

**Table 4 ijerph-18-06228-t004:** The conditional probability of IPD given different configurations of OS, Ethnicity, CCX and Age.

Risk Factor (OS, CCX, Ethnicity, Age)	Probability of Inpatient Mortality
Age < 40	
(1, 2, 2, 1)	0.1375
(1, 2, 1, 1)	0.1376
(2, 1, 1, 1)	0.1387
(1, 1, 1, 1)	0.1391
(2, 2, 1, 1)	0.1395
(1, 1, 2, 1)	0.1401
(2, 1, 2, 1)	0.1407
(2, 2, 2, 1)	0.1417
**Age < 70**	
(1, 2, 2, 3)	0.6755
(1, 2, 1, 3)	0.6756
(1, 1, 1, 3)	0.6761
(1, 1, 2, 3)	0.6764
(2, 1, 2, 3)	0.6927
(2, 1, 1, 3)	0.6944
(2, 2, 1, 3)	0.6944
(2, 2, 2, 3)	0.6946

**Table 5 ijerph-18-06228-t005:** The conditional probability of IPD given different configurations of MADA, Age, MCRP1 and MCRP7.

Risk Factor (MADA, Age, MCRP1, MCRP7)	Probability of Inpatient Mortality
MADA > 35 and Age < 40 years	
(3, 1, 2, 1)—Small CRP Decrease	0
(3, 1, 3, 1)—Large CRP Decrease	0
(3, 1, 1, 2)—Small CRP Increase	0
(3, 1, 1, 3)—Large CRP Increase	0
**MADA > 35 and Age < 70 years**	
(3, 3, 2, 1)—Small CRP Decrease	0.418
(3, 3, 3, 1)—Large CRP Decrease	0.416
(3, 3, 1, 2)—Small CRP Increase	0.496
(3, 3, 1, 3)—Large CRP Increase	0.812
**MADA < 30 and Age < 70 years**	
(1, 3, 2, 1)—Small CRP Decrease	0.515
(1, 3, 3, 1)—Large CRP Decrease	0.513
(1, 3, 1, 2)—Small CRP Increase	0.734
(1, 3, 1, 3)—Large CRP Increase	0.865

**Table 6 ijerph-18-06228-t006:** Summary of the predictive performance results of the BN model developed to model IPD as Illustrated in Figure 7.

Predictive Performance Metric	PPV	NPV	Specificity	Sensitivity	Overall Accuracy	F1-Score
BN for IPD	82%	67.86%	82.6%	85.7%	84.1%	83.7%

**Table 7 ijerph-18-06228-t007:** The heat-mapped, conditional probabilities of ADT given different configurations of Obesity, MADA and MCRPI.

Probability of ADT Given Obesity, MADA and MCRP7	MADA (3) and MCRP1 (1)	MADA (3) and MCRP1 (3)	MADA (1) and MCRP1 (1)	MADA (1) and MCRP1 (3)
**BMI < 30 (Non-Obese patients)**				
<1 day	71.2%	68.7%	10.5%	10.5%
>2 days but <3 days	23.7%	25%	32.7%	30.3%
>3 days	5.1%	6.3%	56.8%	59.2%
**BMI > 30 (Obese patients)**				
<1 day	54.4%	49.4%	13.2%	11%
>2 days but <3 days	40.7%	45.2%	34.6%	32.5%
>3 days	4.8%	5.4%	52.2%	56.5%

**Table 8 ijerph-18-06228-t008:** Summary of the predictive performance results of BN model developed to model ADT.

Predictive Performance Metrics of ADT Category	Balanced Accuracy	Sensitivity (Recall)	Specificity	Precision	Overall Accuracy	F1-Score
<1 day	74.8%	66.2%	83.5%	73.7%	61.5%	69.8%
>2 days but <3 days	60.6%	41.4%	79.8%	53.5%	61.5%	46.7%
>3 days	71.3%	76.9%	65.7%	57.9%	61.5%	66.1%

**Table 9 ijerph-18-06228-t009:** The heat-mapped, conditional probabilities of MOoVS given the different configurations of MADA, OS and MCRP11.

Probability of MOoVS Given Category of OS, MADA and MCRP11	OS (1), MADA (3) and MCRP11 (1)	OS (1), MADA (1) and MCRP11 (2)	OS (1), MADA (1) and MCRP11 (1)	OS (2), MADA (3) and MCRP11 (1)	OS (2), MADA (1) and MCRP11 (1)	OS (2), MADA (1) and MCRP11 (2)
NHF	72.80%	39.30%	34.80%	20.10%	9.80%	1.80%
HF	12.50%	10.10%	38.90%	45.90%	25.40%	12.40%
CPN	14.70%	26.90%	18.60%	34%	39.30%	36.90%
ITU	0%	23.70%	7.70%	0%	25.50%	48.90%

**Table 10 ijerph-18-06228-t010:** The heat-mapped, conditional probabilities of MOoVS given the different configurations of Age, MADA, OS and MCRP11.

Probability of MOoVS Given OS, MADA, MCRP11 and Age	OS (1), MADA (3), MCRP11 (1) and Age (1)	OS (1), MADA (1), MCRP11 (1) and Age (1)	OS (1), MADA (3), MCRP11 (1) and Age (3)	OS (1), MADA (1), MCRP11 (1) and Age (3)	OS (2), MADA (1), MCRP11 (2) and Age (1)	OS (2), MADA (1), MCRP11 (2), and Age (3)
NHF	92.60%	79.80%	61%	39.90%	0%	2.10%
HF	7.40%	20.20%	33.60%	53.90%	0%	22%
CPN	0%	0%	5.40%	3.10%	36.50%	75.90%
ITU	0%	0%	0%	3.10%	63.50%	0%

**Table 11 ijerph-18-06228-t011:** Summary of the predictive performance results of BN developed to model MOoVS.

Predictive Performance Metrics	Balanced Accuracy	Recall (Sensitivity)	Specificity	Precision	Overall Accuracy	F1-Score
NHF	72.3%	56%	88.7%	70.8%	60.25 %	62.5%
HF	68.5%	61.2%	75.8%	51.3%	60.25 %	55.8%
CPN	63.6%	36.2%	91.1%	64.8%	60.25%	66.4%
ITU	80.7%	88.84%	72.7%	59.9%	60.25 %	71.5%

**Table 12 ijerph-18-06228-t012:** The heat-mapped, conditional probabilities of NCPE given the different states of MDD, MADA and UoB. The results suggest that the presence of NCPE is more significantly influenced by the presence of bilateral ground-glass or consolidative CT scan changes.

**Probability of NCPE Given UoB and MDD**	**Bilateral CT Changes and MDD < 400**	**Bilateral CT Changes and MDD > 400**	**Unilateral CT Changes and MDD < 400**	**Unilateral CT Changes and MDD > 400**
**No NCPE**	27.20%	27.80%	86.50%	85.90%
**NCPE**	72.80%	72.20%	13.50%	14.10%
**Probability of NCPE Given Categories of UoB and MADA**	**Bilateral CT Changes and MADA < 30**	**Bilateral CT Changes and MADA > 35**	**Unilateral CT Changes and MADA < 30**	**Unilateral CT Changes** **and MADA > 35**
**No NCPE**	27%	51%	85.30%	98.60%
**NCP**	73%	49%	14.70%	1.40%

**Table 13 ijerph-18-06228-t013:** The heat-mapped, conditional probabilities of NCPE given the different states of MADA, MCRP7 and Obesity.

Probability of NCPE Given Categories of MADA, MCRP7 and Obesity	MADA (1) and MCRP7 (1)	MADA (1) and MCRP7 (3)	MADA (3) and MCRP7 (1)	MADA (3) and MCRP7 (3)
**BMI < 30 (Non-Obese Patients)**				
**No NCPE**	44.20%	30.90%	63%	44.70%
**NCPE**	55.80%	69.10%	37%	55.30%
**BMI > 30 (Obese Patients)**				
**No NCPE**	68.20%	51.50%	94.30%	87.80%
**NCPE**	31.80%	48.50%	5.70%	12.20%

**Table 14 ijerph-18-06228-t014:** Summary of predictive performance results of the BN learned for “NCPE”. The computed F1-score of almost 86% shows the classification prediction of the learned BN for NCPE is precise and robust.

Predictive Performance Metric	PPV	NPV	Specificity	Sensitivity	Overall Accuracy	F1-Score
BN for IPD	83.7%	80.9%	75%	87.9%	82.7%	85.8%

## Data Availability

Restrictions apply to the availability of the anonymised data used in this paper. Data was obtained from the Milton Keynes hospital electronic patient records, in accordance with good clinical practice guidelines, after obtaining health research authority (HRA) approval. The datasets are available from the authors with the permission of the Milton Keynes hospital authorities.

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
