# Peer review of "Using Machine Learning Algorithms to Develop a Clinical Decision-Making Tool for COVID-19 Inpatients"

_ijerph, 2021, doi:10.3390/ijerph18126228_

Round 1

Reviewer 1 Report

>> General appreciation.

- The introduction section is well presented. The introduction explains the general context of the problem and the many facets of the study. Provide bibliographic references to define the topics.
- The main text of the research is clear and easy to understand.
- The discussion presents relevant aspects of the study.
- The conclusions are refined.
- Provide bibliographic references to define the topics mentioned.
- Points to improve:
-    This research is partially reproducible according to the unavailability of data and source code.

A detail of the review is shown below:

>> Introduction

Does the introduction provide sufficient background information for readers not in the immediate field to understand the problem/hypotheses?
-    Yes, the introduction is appropriated.

Are the reasons for performing the study clearly defined?
-    Yes.

Are the study objectives clearly defined?
-    Yes.

>> Methods/technical rigor

Are the methods used appropriate to the aims of the study?
-    Yes, the methods are appropriate.

Is sufficient information provided for a capable research to reproduce the experiments described?
-    The methods are provided to carry out a reproduction of the experiment.
-    This research is partially reproducible according to the unavailability of data and source code.
-    The research does not include the source code of the described functions. Please add the source code in the annexe section if it is small, or relevant code fragments of the proposed design, especially from point 2.6.2. Balancing Outcomes and 2.6.3. Feature selection, which is the initial basis of the model. This topic is explicitly detailed in the instructions: “Computer Code and Software - For work where novel computer code was developed, authors should release the code either by depositing in a recognized, public repository such as GitHub or uploading as supplementary information to the publication. The name, version, corporation and location information for all software used should be clearly indicated. Please include all the parameters used to run software/programs analyses” (see https://www.mdpi.com/journal/remotesensing/instructions#suppmaterials).
-    In the "Data Availability Statement" section, you should only indicate whether the data is available or not. Indicating that the data is anonymized or not is not relevant and that explanation of the article is already made. The objective of the section is information on whether the data can be obtained or not.  Format: “The data are not publicly available due to [insert reason here]” (see https://www.mdpi.com/journal/remotesensing/instructions#suppmaterials).
-    I have added the following statement to the publisher because the peer review must be awaited and the publisher's guidelines verified.: “Any restrictions on sharing must be discussed with the editor at submission who reserves the right to decline the study if these conditions are found to be unduly prohibitive. “

Are any additional experiments required to validate the results of those that were performed?
-    No.

Are there any additional experiments that would greatly improve the quality of this paper?
-    No.

Are appropriate references cited where previously established methods are used?
-    Yes

>>> Results/Statistics

Are the results clearly explained and presented in an appropriate format?
-    Yes.

Do the figures and tables show essential data or are there any that could easily be summarized in the text?
-    Not applied.

Are any of the data duplicated in the graphics and/or text?
-    No

Are the figures and tables easy to interpret?
-    Partially.
-    Figure 1a-d is unreadable due to font size. Also, the lines are blurry.
-    Figure 4 is unreadable due to font size.
-    I see that some figures and texts are blurred, please consider re-importing them to svg, eps or pdf and attach them in your latex folder. The diagrams explain the result of the experiment very well, consider that it is one of the things that are read first when making a first reading of the article.

Are there any additional graphics that would add clarity to the text?
-    The figures contain the information summarized and clear in content.

Have appropriate statistical methods been used to test the significance of the results?
-    It’s not necessary.

>> Discussion

Are all possible interpretations of the data considered or are there alternative hypotheses that are consistent with the available data?
-    No applied.

Are the findings properly described in the context of the published literature?
-    Yes.

Are the limitations of the study discussed? If not, what are the major limitations that should be discussed?
-The limitations are explicitly defined.

>>> Conclusions

Are the conclusions of the study supported by appropriate evidence or are the claims exaggerated?
-    The conclusions are balanced and mention the relevant aspects of the research.

>> Literature cited (introduction, results, discussion)

Is the literature cited balanced or are there important studies not cited, or other studies disproportionately cited?
-    Yes.

Please identify statements that are missing any citations, or that have an insufficient number of citations, given the strength of the claim made.
-    The number of citations is appropriate

>>> Significance and Novelty

Are the claims in the paper sufficiently novel to warrant publication?
-    Yes.

Does the study represent a conceptual advance over previously published work?
-    Yes.

Author Response

Please see the attached file or my replies below:

- The introduction section is well presented. The introduction explains the general context of the problem and the many facets of the study. Provide bibliographic references to define the topics.
- The main text of the research is clear and easy to understand.
- The discussion presents relevant aspects of the study.
- The conclusions are refined.
- Provide bibliographic references to define the topics mentioned.
- Points to improve:
-    This research is partially reproducible according to the unavailability of data and source code.

Response:  Restrictions apply to the availability of the anonymized data used in this paper. Data was ob-tained from the Milton-Keynes hospital electronic patient records, in accordance with good clini-cal practice guidelines, after obtaining health research authority (HRA) approval. The datasets are available from the authors with the permission of the Milton Keynes hospital authorities. However, the data is not publicly available, but the authors are happy to share their codes written in R package, upon request from the corresponding authors.

The authors are creating a structured GitHub to include all R codes used for the different tasks addressed in this paper, after they have annotated each line of the codes, which will be publicly available very soon.

A detail of the review is shown below:

>> Introduction

Does the introduction provide sufficient background information for readers not in the immediate field to understand the problem/hypotheses?
-    Yes, the introduction is appropriated.

Are the reasons for performing the study clearly defined?
-    Yes.

Are the study objectives clearly defined?
-    Yes.

Response:   Thank you for approving the above elements.

>> Methods/technical rigor

Are the methods used appropriate to the aims of the study?
-    Yes, the methods are appropriate.

Is sufficient information provided for a capable research to reproduce the experiments described?
-    The methods are provided to carry out a reproduction of the experiment.
-    This research is partially reproducible according to the unavailability of data and source code.
-    The research does not include the source code of the described functions. Please add the source code in the annexe section if it is small, or relevant code fragments of the proposed design, especially from point 2.6.2. Balancing Outcomes and 2.6.3. Feature selection, which is the initial basis of the model.

Response:  Restrictions apply to the availability of the anonymized data used in this paper. Data was ob-tained from the Milton-Keynes hospital electronic patient records, in accordance with good clini-cal practice guidelines, after obtaining health research authority (HRA) approval. The datasets are available from the authors with the permission of the Milton Keynes hospital authorities. However, the data is not publicly available, but the authors are happy to share their codes written in R package, upon request.

This topic is explicitly detailed in the instructions: “Computer Code and Software - For work where novel computer code was developed, authors should release the code either by depositing in a recognized, public repository such as GitHub or uploading as supplementary information to the publication. The name, version, corporation and location information for all software used should be clearly indicated. Please include all the parameters used to run software/programs analyses” (see https://www.mdpi.com/journal/remotesensing/instructions#suppmaterials).

Response:  Restrictions apply to the availability of the anonymized data used in this paper. Data was ob-tained from the Milton-Keynes hospital electronic patient records, in accordance with good clini-cal practice guidelines, after obtaining health research authority (HRA) approval. The datasets are available from the authors with the permission of the Milton Keynes hospital authorities. However, the data is not publicly available, but the authors are happy to share their codes written in R package, upon request.

-    In the "Data Availability Statement" section, you should only indicate whether the data is available or not. Indicating that the data is anonymized or not is not relevant and that explanation of the article is already made. The objective of the section is information on whether the data can be obtained or not.  Format: “The data are not publicly available due to [insert reason here]” (see https://www.mdpi.com/journal/remotesensing/instructions#suppmaterials).

Response:   Agreed and “Data Availability Statement” section was revised as follows:

Restrictions apply to the availability of the anonymized data used in this paper. Data was obtained from the Milton-Keynes hospital electronic patient records, in accordance with good clinical practice guidelines, after obtaining health research authority (HRA) approval. The datasets are available from the authors with the permission of the Milton Keynes hospital authorities.

-    I have added the following statement to the publisher because the peer review must be awaited and the publisher's guidelines verified.: “Any restrictions on sharing must be discussed with the editor at submission who reserves the right to decline the study if these conditions are found to be unduly prohibitive. “

Response:   Thank you for pointing us to the instruction regarding sharing the data and codes. We are happy to share the codes, and revised our statement regarding sharing the data based on the provided instructions as above.

Are any additional experiments required to validate the results of those that were performed?
-    No.

Are there any additional experiments that would greatly improve the quality of this paper?
-    No.

Are appropriate references cited where previously established methods are used?
-    Yes

>>> Results/Statistics

Are the results clearly explained and presented in an appropriate format?
-    Yes.

Do the figures and tables show essential data or are there any that could easily be summarized in the text?
-    Not applied.

Are any of the data duplicated in the graphics and/or text?
-    No

Are the figures and tables easy to interpret?
-    Partially.
-    Figure 1a-d is unreadable due to font size. Also, the lines are blurry.

Response: We have increased the resolutions of these images, by representing each figure separately. However, in order to keep the text as it is, we refer to these 4 plots as Figure 1a to Figure 1d.

-    Figure 4 is unreadable due to font size.

Response: We have increased the resolution of this figure, by re-running it in R code. 

-    I see that some figures and texts are blurred, please consider re-importing them to svg, eps or pdf and attach them in your latex folder. The diagrams explain the result of the experiment very well, consider that it is one of the things that are read first when making a first reading of the article.

Response: We have increased the resolutions of the figures given in the supplementary file, by re-running them in R code. You could observe that the figures now have much better resolution. 

Are there any additional graphics that would add clarity to the text?
-    The figures contain the information summarized and clear in content.

Have appropriate statistical methods been used to test the significance of the results?
-    It’s not necessary.

>> Discussion

Are all possible interpretations of the data considered or are there alternative hypotheses that are consistent with the available data?
-    No applied.

Are the findings properly described in the context of the published literature?
-    Yes.

Are the limitations of the study discussed? If not, what are the major limitations that should be discussed?
-The limitations are explicitly defined.

>>> Conclusions

Are the conclusions of the study supported by appropriate evidence or are the claims exaggerated?
-    The conclusions are balanced and mention the relevant aspects of the research.

>> Literature cited (introduction, results, discussion)

Is the literature cited balanced or are there important studies not cited, or other studies disproportionately cited?
-    Yes.

Please identify statements that are missing any citations, or that have an insufficient number of citations, given the strength of the claim made.
-    The number of citations is appropriate

>>> Significance and Novelty

Are the claims in the paper sufficiently novel to warrant publication?
-    Yes.

Does the study represent a conceptual advance over previously published work?
-    Yes.

The authors would like to thank reviewer’ comments for their invaluable comments and the opportunity they provided us to improve our paper and make this manuscript closer to publication in the International Journal of Environmental Research and Public Health.

Reviewer 2 Report

In this article, the authors develop a reliable, multivariable predictive model for COVID-19 in-patient outcomes to aid risk-stratification and earlier clinical decision-making. 
The authors deal with an issue that concerns all citizens of this world.
In this one year, scientists from all over the world are trying to decode COVID-19.
The article has a perfect structure. The sections have a good flow.
The technical contribution of this article is limited, but the results are reliable and can be useful for further clinical researches.
The authors have described the methodology of their models and have cited the required references.
References are adequate and up-to-date.
The authors have conducted many experiments to prove their approach in terms of accuracy. 
The results have enough commentary and documentation.
I have to point out the following weaknesses that need improvement, further explanation and correction.
1)Authors' English is not up to standard. There are many syntax, grammar and punctuation errors that need to be corrected.
2)The models need to be explained with more details from ML technical points of view. Why were machine learning methods selected for the task of interest? What is its superiority?
3)There is a  limitation in this article. It is the limited training data. Could you comment on this aspect? This calls into question your approach?
4)Could the authors mention the implementation environment of their experiments?
5)Ιn the conclusion section, the authors should summarize the superiority of their methods (and numerically).

Author Response

Please find below my responses to the respected Reviewer 2's comments, which are also included in the attached cover letter:

Comments and Suggestions for Authors

In this article, the authors develop a reliable, multivariable predictive model for COVID-19 in-patient outcomes to aid risk-stratification and earlier clinical decision-making. 
The authors deal with an issue that concerns all citizens of this world.
In this one year, scientists from all over the world are trying to decode COVID-19.
The article has a perfect structure. The sections have a good flow.
The technical contribution of this article is limited, but the results are reliable and can be useful for further clinical researches.
The authors have described the methodology of their models and have cited the required references.
References are adequate and up-to-date.
The authors have conducted many experiments to prove their approach in terms of accuracy. 
The results have enough commentary and documentation.

Response: Thank you for acknowledging strengths in our manuscript. We have addressed your constructive suggestions, which are listed below.

I have to point out the following weaknesses that need improvement, further explanation and correction.
1) Authors' English is not up to standard. There are many syntax, grammar and punctuation errors that need to be corrected.

Response: We have thoroughly proofread the entire paper, and corrected the syntax, grammar and punctuation errors that we have found in the paper. Thank you.

2) The models need to be explained with more details from ML technical points of view. Why were machine learning methods selected for the task of interest? What is its superiority?

Response: Thank you for pointing out this comment. We agree that the motivations behind using the BNs as the main ML method was not well discussed in “Introduction”. As a result, we have added the following section in “Introduction” to explain why the BNs as the main ML method was selected in this study. 

The multivariate predictive model showcased in this study uses Bayesian Networks (BNs) which have received increasing attention during the last two decades (McLach-lan et al., 2020; Kyrimi et al., 2019) for their efficacy in tackling challenging and com-plex problems whilst also aiding in making decisions under uncertainty (Constantinou et al., 2018). The ever-increasing volumes of health data has created potential for developing new knowledge that could improve clinical practise and patient care. The BNs and other machine learning (ML) methods have been extensively utilised in a di-verse range of health topics from genomics (Libbrecht and Noble, 2015; Chatrabgoun et al., 2018, 2020) to treatment selection, and outcome, prognosis, prediction (Parmar et al., 2015). A compelling advantage of BNs over other suitable data-driven methods is that they do not explicitly need massively large datasets. Furthermore, BNs can combine the elicited knowledge of experts in circumstances where data are limited, and still produce meaningful and accurate decision-support systems (O’Hagan et al., 2006; Daneshkhah and Oakley, 2010; Daneshkhah et al., 2017; Smith and Daneshkhah, 2010).

This paper seeks to develop a quantitative tool to aid risk-stratification, and earlier clinical decision-making for adult COVID-19 inpatients by benefiting from the proper-ties of BNs, including making reliable predictions, and being robust in making decisions under various sources of uncertainties in data.

It should be noted that we have used several other ML methods, including “multiple imputation technique” to tackle the missing values, SMOTE to overcome the imbalance in the dataset, and recursive feature elimination (RFE) method using the random forest function, which all have discussed in details in Section 2.6.

3) There is a limitation in this article. It is the limited training data. Could you comment on this aspect? This calls into question your approach?

Response: Thank you for asking this question. Indeed one of the limitation of this study was the limited training data. This is mainly due to various reasons including lack of access to enough data when this research was initially started, and other health ethical regulations which inhibited us to gather more data. As a result, we have selected a more efficient and robust ML method, that is, BNs, which do not explicitly need massively large datasets, and have this nice property that they model structure and predictions can be efficiently updated in the light of the information gathered in the form of expert judgments. In the discussion, added to Introduction (as a response to your 2nd question), we have addressed this point.

This limitation was also critically discussed in the last paragraph of “Section 4. Discussion”.  

In addition, we have already considered the possible development of the developed models in the paper if more training data was provided. Please see below (as discussed in “Conclusions”):

The models, if provided with more training data, have the potential to be refined even further. Future research is required to externally validate our models and demonstrate their utility as clinical decision-making tools.

4) Could the authors mention the implementation environment of their experiments?

Response: All data analysis and results reported in this paper were implemented in R package. However, the anonymized data is not publicly available (due to UK health research authority (HRA) regulations), but the authors intend to share their codes written in R package, for the selected sections in the supplementary file attached to this paper.

5) In the conclusion section, the authors should summarize the superiority of their methods (and numerically).

Response: We have added the following discussion to the Conclusion:

Our model not only computes the probability distributions of children nodes given the values of their parent nodes, but also the distributions of the parents given the values of their children. In other words, they can proceed not only from causes to consequences, but also deduce the probabilities of different causes given the consequences. All these probabilities can be instantly computed using the codes, developed in R and are in accessible, upon request from the corresponding authors.

In addition, we have added the following paragraph at the start of the Conclusion:

In this study, we were facing several numerical challenges whilst constructing a ro-bust, reliable and computationally efficient probabilistic data-driven model, including a very small sample size in comparison to the dimension of input variables, and significant rates of missing values for several variables. These challenges were all efficiently resolved by selecting and employing a range of ML methods. The output of this study was a quantitative tool, which can aid in both risk-stratification, and earlier clinical decision making, for COVID-19 inpatients.

The authors would like to thank reviewer’ comments for their invaluable comments and the opportunity they provided us to improve our paper and make this manuscript closer to publication in the International Journal of Environmental Research and Public Health.

Reviewer 3 Report

In this work, the authors use Bayesian networks to analyze and predict 4 outcomes for COVID-19 patients. These 4 outcomes are: (i) inpatient mortality (IPD), (ii) duration of COVID-19 treatment (ADT), (iii) maximum level of oxygen or ventilatory support during inpatient stay (MOoVS),  and (iv) confirmed diagnosis of pulmonary embolism during inpatient stay (NCPE). To perform this study the Bayesian networks are learned from a dataset and after that, the networks are refined by experts. 

The article is well written and the method presented seems pretty interesting and potentially effective. The analysis of the results seems rigorous and well-founded. However, this reviewer has some reservations about the methodology carried out:

-It seems that the preprocessing steps are made using the whole dataset. Some preprocessing techniques could bias the results since the information in the test set is used for training. For example, using the test samples in SMOTE invalidate the obtained results because those test samples (with modifications) would be included in the training set. Therefore, the preprocessing steps should be made only in the training data. That said, preprocessing can be performed on the training data and compute the performance measures using the test set, but to analyze the results from the Bayesian networks, the complete dataset can be used.

-The validation method used to compute the performance measures, should be mentioned. 

-The AUC measure could be considered as a convenient performance measure on these grounds, where the different types of errors have different costs and it is insensitive to imbalanced classes.

-The search method and the metric used for the Bayesian networks should explicitly be mentioned guaranteeing the reproducibility of the results. 

-To analyze the results obtained from the Bayesian networks, the study should circumscribe to the Markov blanket of the outcome studied. 

On the other hand, as long as the authors have the intention to continue this research with more patient data, the actual dataset should be publicly available (reproducible research).

Author Response

Please find below my responses to the respected Reviewer 2's comments, which are also included in the attached cover letter:

In this work, the authors use Bayesian networks to analyze and predict 4 outcomes for COVID-19 patients. These 4 outcomes are: (i) inpatient mortality (IPD), (ii) duration of COVID-19 treatment (ADT), (iii) maximum level of oxygen or ventilatory support during inpatient stay (MOoVS),  and (iv) confirmed diagnosis of pulmonary embolism during inpatient stay (NCPE). To perform this study the Bayesian networks are learned from a dataset and after that, the networks are refined by experts. 

The article is well written and the method presented seems pretty interesting and potentially effective. The analysis of the results seems rigorous and well-founded.

Response: Thank you for acknowledging strengths in our manuscript. We have addressed your constructive suggestions, which are listed below.

However, this reviewer has some reservations about the methodology carried out:

-It seems that the preprocessing steps are made using the whole dataset. Some preprocessing techniques could bias the results since the information in the test set is used for training. For example, using the test samples in SMOTE invalidate the obtained results because those test samples (with modifications) would be included in the training set. Therefore, the preprocessing steps should be made only in the training data. That said, preprocessing can be performed on the training data and compute the performance measures using the test set, but to analyze the results from the Bayesian networks, the complete dataset can be used.

Response: Thank you for pointing out this point.

There are conflicting views on how the SMOTH must be implemented, whether on the original data, or only on the training dataset. There are several studies (including [26-29], and also, Shatnawi and Al-Sharif, 2012; Flores and Gomes, 2015; and Agrawal et al., 2017) that recommend applying the SMOTE on the entire dataset, and reported that SMOTE is better than the original data for nine classifiers, including Bayesian network. In addition, the minority class percentages of two of responses (NCPE and MOoVS) were less than 5%, which almost made it impractical to create reliable training/testing datasets prior to applying the SMOTE on the training datasets. Nevertheless, the authors agree with the recommendations, that further investigation should be conducted to explore the validity of the proposed models when the SMOTE applied on training dataset rather than the whole dataset. As a result, we have added the following explanation to Discussion, as future work:   

As a future work, it would be appealing to further investigate the impact of SMOTH technique on the validity of the BN classifiers. As discussed above, it was observed that the response variables in this study are significantly imbalance (e.g., the minority class percentages of NCPE or MOoVS were less than 5%). In order to overcome the imbalance in the dataset, the SMOTE technique was applied on the entire dataset, as recommended in [26–29]. However, the alternative approach which is more plausible would be to apply the SMOTH on training datasets only, and re-evaluate the performance measures using the test data. 

Additional References (that were not added to the paper):

-The validation method used to compute the performance measures, should be mentioned. 

Response: Thank you for pointing out this point. We added the following explanation in Section 2, and before illustrating Figure 2:

In particular, the BN shown in Figure 2 was selected by employing hill-climbing (or hc) algorithm and benchmarked with other suitable learning algorithms (e.g., Tabu Search or simply tabu) available in “bnlearn” library in R package. We then used the cross-validation, which is a standard way to obtain unbiased estimates of a model's goodness of fit to select the best models out of the learned networks using the learning algorithms mentioned above. The cross-validation method used in this paper is 5-fold cross-validation that can be simply computed using “bn.cv” function in “bnlearn” package. This function provides us with Log-Likelihood Loss, its standard deviation and BIC. Both “tabu” and “hc” algorithms suggested the network structure illustrated in Figure 2, as the best network learn from data, with the same BIC.   

And also added the following sentence at the end of the first paragraph of page 11 (lines 293-297)

The computed BIC metric for the network shown in Figure 3 (-4190.4, equivalent to 9.03 of Log-Likelihood loss value and 0.008 standard deviation of the loss) is smaller than BIC of the model shown in Figure 2, which was computed to be, -4000.3 (equivalent to 8.69 of Log-Likelihood loss value and 0.011 standard deviation of the loss).

We have used similar methods for validating the best BNs selected for other outcomes.  

-The AUC measure could be considered as a convenient performance measure on these grounds, where the different types of errors have different costs and it is insensitive to imbalanced classes.

Response: We totally agree with the reviewer another commonly used metric to evaluate the model prediction performance is the ROC, which plots “sensitivity” as a function of “1-specificity”. In addition, AUC can be also considered as a metric to judge overall performance of classification models (Hand, 1997). The only issue with using AUC is that it is more effective for binary classification, while two outcomes addressed in this paper are multi-class variables. As a results, and in order to be consistent, we have used k-fold cross validation and Schwarz’ Bayesian information criterion (or simply BIC) as the means to examine the model performance.

-The search method and the metric used for the Bayesian networks should explicitly be mentioned guaranteeing the reproducibility of the results. 

Response: We added the following explanation in Section 2, and before illustrating Figure 2. We used hill-climbing (or hc) and Tabu search (or simply tabu) algorithms as the search method to selected the best network structure for each outcome based on the data only. The full description is given below:

In particular, the BN shown in Figure 2 was selected by employing hill-climbing (or hc) algorithm and benchmarked with other suitable learning algorithms (e.g., Tabu Search or simply tabu) available in “bnlearn” library in R package. We then used the cross-validation, which is a standard way to obtain unbiased estimates of a model's goodness of fit to select the best models out of the learned networks using the learning algorithms mentioned above. The cross-validation method used in this paper is 5-fold cross-validation that can be simply computed using “bn.cv” function in “bnlearn” package. This function provides us with Log-Likelihood Loss, its standard deviation and BIC. Both “tabu” and “hc” algorithms suggested the network structure illustrated in Figure 2, as the best network learn from data, with the same BIC.   

-To analyze the results obtained from the Bayesian networks, the study should circumscribe to the Markov blanket of the outcome studied. 

Response: Thank you for this remark, but we believe going in depth to link the derive results for the presented Bayesian networks with the Markov blanket could make the paper very technical and unnecessary long.

On the other hand, as long as the authors have the intention to continue this research with more patient data, the actual dataset should be publicly available (reproducible research).

Response:  Restrictions apply to the availability of the anonymized data used in this paper. Data was ob-tained from the Milton-Keynes hospital electronic patient records, in accordance with good clini-cal practice guidelines, after obtaining health research authority (HRA) approval. The datasets are available from the authors with the permission of the Milton Keynes hospital authorities. However, the data is not publicly available, but the authors are happy to share their codes written in R package, upon request.

The authors are creating a structured GitHub to include all R codes used for the different tasks addressed in this paper, after they have annotated each line of the codes, which will be publicly available very soon.

The authors would like to thank reviewer’ comments for their invaluable comments and the opportunity they provided us to improve our paper and make this manuscript closer to publication in the International Journal of Environmental Research and Public Health.

Round 2

Reviewer 2 Report

I have no additional remarks on the revised version.

The authors have addressed my concerns.